# A qualitative study to assess perceptions, barriers, and motivators supporting smokeless tobacco cessation in the US fire service

**Nattinee Jitnarin**[1]*, **Walker S. C. Poston**[1], **Sara A. Jahnke**[1], **Christopher K. Haddock**[1], **Hannah N. Kelley**[1], **Herbert H. Severson**[2]

**1** Center for Fire, Rescue & EMS Health Research, NDRI–USA, Inc., Leawood, Kansas, United States of America, **2** Oregon Research Institute, Eugene, Oregon, United States of America

* jitnarin@ndri-usa.org

## Abstract

While firefighters currently have low smoking rates, rates of smokeless tobacco (SLT) use among this population are remarkably high and substantially greater than similar occupational groups, and the general population. This study explored determinants associated with SLT use, barriers to cessation, and motivators for SLT cessation in the fire service. Key informant interviews were conducted in 23 career firefighters who were current (n = 14) and former (n = 9) SLT users from across the U.S. Discussions were recorded and independently coded according to questions and themes. Major themes that developed among firefighters regarding SLT use determinants included positive perceptions of SLT products, social influences from their peers and family members, acceptability of SLT use in the fire service, and a coping resource for job stress. Firefighters discussed several barriers to SLT cessation, including intrapersonal barriers such as SLT use habits and its dependency, concerns about withdrawal symptoms; and social-environmental barriers including lack of support from health and other services providers, and lack of enforcement of existing tobacco policies regarding SLT use. Firefighters also mentioned both internal and external motivators for cessation. Internal motivators included self-motivation and their health concerns while external motivators included friends and family support, incentives or rewards, and price of SLT products. Findings provide unique perspectives from firefighters on factors that influence SLT use and barriers and motivators to SLT cessation. These are insufficiently assessed and considered by the fire service organizations and their health care providers. Thus, the organizations must understand these issues in order to mitigate barriers and motivate the personnel to quit using SLT. Information gained from firefighters who were current and former SLT users can be used to develop an effective, culturally-tailored intervention that is acceptable to fire service personnel.

**Data Availability Statement:** Data cannot be shared publicly because participant consent was not given for interview transcripts to be shared.

However, requests for anonymized data can be made to the Assistant Grants Management Director Carol Tarzian via email (tarzian@ndri-usa.org).

**Funding:** This study was supported by a grant 129326-MRSG-16-104-01-CPPB grant from the American Cancer Society, (www.cancer.org) to NJ. The funder had no role in study design, data collection and analysis, decision to publish, or preparation of the manuscript.

**Competing interests:** The authors have declared that no competing interests exist.

## Introduction

Firefighters are charged with protecting the citizens and property in the communities they serve. The nature of the profession makes firefighting a physically and mentally demanding occupation. Firefighters must respond to emergencies at a moment's notice and often face dangerous and challenging work conditions. The occupational exposures from required duties put firefighters at risk for both injuries and diseases, making the health and wellness of the population a key concern [1–3]. Occupational hazards specific to the nature of the Fire Service are hypothesized to play a role in the increased risk of cancers. For instance, exposure to recognized or probable carcinogens, such as benzene, polycyclic aromatic hydrocarbons, and diesel fumes is common on the fire ground and during fire combustion. Because of these factors, firefighters experience a greater risk and high proportionate mortality ratio of several forms of cancer [4–7].

Due to the physically and mentally demanding nature of the profession, there has been a strong emphasis on health promotion in the fire service, including encouraging firefighters to be tobacco-free. For example, the Fire Service Joint Labor Management Wellness Fitness Initiative, the model "gold standard" wellness program for the fire service, suggests that all departments adopt the Tobacco Cessation Policy [8]. This policy recommends fire service personnel should not use tobacco products inside the work-site, within or on the fire department apparatus, or inside training facilities. The National Fire Protection Association Standard on Fire Department Occupational Safety and Health Programs (NFPA 1500) requires all fire department facilities be designated smoke-free [9]. In addition, the labor union representing firefighters, the International Association of Fire Fighters, formed a partnership with Pfizer, a pharmaceutical company, to create the "Campaign for a Smoke-Free Union" which includes advice and help on reducing tobacco use, including smoking cessation [10].

Given the fire service's stance and efforts to address tobacco use, career firefighters' cigarette smoking prevalence is relatively low (13.2%) and comparable to the general public (15.6%) [11] and related occupational groups such as military personnel (14.4%) [12]. However, the prevalence of SLT use in the fire service is very high. For example, we found that between 10.5%-18.4% of male firefighters exclusively used SLT [13, 14], which is almost three times the rate of SLT use found in the general population [11]. Firefighters report rates comparable to other occupational groups in the high-end range of use. For instance, the rate for construction workers was 7.9%-10.9%, while the rate of agriculture workers and mining workers were 8.8%-15.8% and 18.8%-28.0%, respectively [15, 16].

While many cancer-related occupational hazards are unavoidable for firefighters, it is critical that modifiable risk factors for cancer be identified and eliminated. A growing body of literature over the past decade highlights SLT use as an important cancer risk factor. SLT use is likely related to cancer development with more than 30 carcinogens contained in SLT products, including aldehydes, polycyclic aromatic hydrocarbons, and nitrosamines [17, 18]. Data from epidemiological studies suggest that the relative risk of oral and pharyngeal, esophageal, and pancreatic cancers were significantly higher among SLT users compared to those who did not use SLT [19]. A number of studies also have found increased risk of all-cause mortality in SLT users compared to non-tobacco users, ranging from 20–40% excess risk [19–21]. Given the critical role firefighters play in the public safety net, the exceptionally high rates of SLT use among firefighters is an important issue in cancer prevention research.

Despite evidence about the negative health consequences of SLT, and exceptionally high rates of use, no formative research exists which systematically examines the social and cultural norms and practices around firefighters' SLT use. Additionally, no SLT education or treatment programs have been implemented for the fire service, and no occupationally tailored SLT

cessation intervention exists for firefighters. Previous research has found that cessation programs tailored to the unique culture and needs of a population produce significantly better results than generic programs or programs created for other groups [22, 23]. A key first step for developing an effective, culturally-tailored intervention that is acceptable to fire service personnel is to assess perspectives, identify barriers to SLT cessation, determine motivations for quitting, and identify cessation interventions customized to the fire service culture.

Social Cognitive Theory (SCT) has been used in various health promotion approaches, including changing behavior related to substance abuse and tobacco use cessation [24, 25]. SCT posits that behaviors change, and learning often occurs through the belief in one's ability to perform the desired behavior (i.e., self-efficacy) and the perception that the desired behavior will lead to certain consequences (i.e., outcome expectations). In addition, perceived barriers and facilitators to performing actions can influence whether individuals will change their undesirable behaviors. Evidence shows that improving self-efficacy and positive social support significantly impact on individual's achievement and changing their undesirable behaviors [26–28]. It was, therefore, considered beneficial for our study to apply SCT to explore firefighters' views and perceptions about SLT use and cessation and to understand how personal, social, and environmental determinants may influence and affect their SLT use behaviors, allowing for tailoring and development of the SLT cessation programs designed to the needs of firefighters who want to quit. The current study explores firefighters' perspectives about SLT use and cessation based on data from key informant interviews with a national sample of career firefighters who were current and former SLT users.

## Methods

This qualitative study used semi-structured interviews to explore reasons for using SLT and barriers and motivators to quitting in a sample of current and former SLT users among firefighters. The present study received Institutional Review Board (IRB) approval from the National Development and Research Institutes Inc., IRB.

### Participants

Participants were solicited through two methods: 1) contacting participants from our previous studies [13, 29], and 2) posting on the "Secret List"–a popular email listserv in the fire service (see Jitnarin et al., 2020 [30], for additional information on sampling procedures, eligibility criteria, and recruiting). Briefly, participant selection was based on US region (East, Central, West), role in the fire service (e.g., firefighter, captain, fire chief), and SLT use status (current users—those who reported exclusively using SLT on multiple occasions within the past month; and former users—those who reported using SLT regularity in the past, but not currently using SLT or any tobacco products). For this study, only career firefighters were included because their exposure to occupational risk factors for SLT use would be arguably greater than volunteers for whom firefighting is a form of community service. Also, volunteer firefighters have less frequent interactions with their departments and colleagues than career firefighters. In addition, women firefighters were not included in the study due to the very low SLT use prevalence compared with men in the fire service (0.5%; [31]). We determined that thematic saturation (i.e., no new themes emerged) and a sufficiently diverse sample was reached at 23 interviews.

### Interview methodology

Data were collected between March-August 2017 to learn firefighters' opinions about SLT use and cessation and perceived barriers and motivators to quitting. Semi-structured interviews

were conducted using an interview guided designed based on Social Cognitive Theory (SCT) [24] and previous literature regarding SLT use and cessation [32–34]. Questions in the guided interview included a selection of open- and closed-ended questions on SLT use history, factors that reinforced using SLT, motives for quitting, barriers to quitting, knowledge and beliefs regarding SLT use, and SLT cessation services in the fire service organization. These questions were framed to encourage participants to discuss their ideas freely without being led by a fixed format. Thus, they were designed to encourage new and unanticipated information [35]. As participants shared their experiences, probes were used to retrieve more in-depth information about their thought and opinion regarding SLT use. The interview questions reported here are a subset from a larger project about cancer risk and the culture of SLT use in the US fire service. (S1 File).

Participants were scheduled for a telephone interview after agreeing to participate in the study. They also received an email with the Consent Form describing the purpose of the study and the consent process. Prior to the interview, investigators explained the study procedures and asked for participants' verbal consent. Participants also completed a short survey with questions about demographics and occupational history. The interviews were digitally recorded, and field notes were taken. Researchers (NJ, CP, HK) carried out the semi-structured interviews, ranging from 20 to 50 minutes (median 42 minutes). Of the 23 interviewed participants, one participant responded to a fire call, resulting in only 20 minutes of the interview. Issues relating to the confidentiality, anonymity, and voluntary nature of the data collection were emphasized.

## Data analysis

Interview recordings were transcribed verbatim. A two-phase process was used to capture the meaning behind the transcribed text with the overall purpose of understanding major themes across and between transcripts. First, researchers reviewed the transcribed documents to develop a familiarity with the text and began to generate the initial codes with an open coding process [36–38]. This inductive analytic approach allowed researchers to explore the new themes or unanticipated findings that may not emerge when using a deductive method within SCT [39, 40]. The themes were then reviewed and grouped into categories [38]. Next, data were reanalyzed using a deductive approach by identifying passages that exemplified key concepts or ideas related to significant patterns and themes. Domains focused on determinants related to SLT use and cessation were identified using SCT and findings from previous literature and linked to questions in the interview guide.

The transcripts were uploaded to NVivo 12 software [41], a qualitative data analysis program that allows researchers to highlight and code data into "parent" nodes for overall themes and "child" nodes for subthemes. Summaries were then made within each major/parent theme. Data were analyzed separately for each group (current and former SLT users). The use of multiple reviewers assisted in establishing the thematic framework. Two primary coders independently coded all 23 transcripts, compared their analyses, and discussed any discrepancies until consensus was reached. Coding was regularly reviewed and verified by the third researcher to identify and incorporate emergent themes and for data validity. In addition, the third researcher conducted independent coding of a subset of data (n = 5) to ensure that the final coding scheme had adequate reliability.

## Results

The 23 firefighters included in the study (Table 1) were all men, had a mean age of 42.3 years (SD = 7.2). Most were White (91.3%), non-Hispanic/Latino (87.0%), and served in Fire Chief

**Table 1. Demographic characteristics of participants (N = 23).**

| Characteristics | Former Users (n = 9) | Current Users (n = 14) |
|---|---|---|
| Age years (SD; Range) | 44.3 (6.9; 34–54) | 40.9 (7.4; 30–56) |
| Race (%) | | |
| White | 88.9 | 92.9 |
| Of Hispanic origin (% yes) | 22.2 | 7.1 |
| Time in the fire service (years; Range) | 23.3 (11–34) | 19.5 (10–37) |
| Rank/position in fire department (%) | | |
| Any Firefighter | 22.2 | 35.6 |
| Any Officer | 22.2 | 28.5 |
| Any Fire Chief | 55.6 | 35.9 |

rank (43.5%). Fourteen (or 60.9%) and nine (39.1%) participants were current and former SLT users, respectively.

Key qualitative findings are described below and organized according to the major themes associated with SLT use and expressed across all participants. An overview of SLT use determinants, barriers to cessation, and motivators for cessation identified from the interview is presented in Table 2. Illustrative comments from interviewed participants are included as appropriate.

## Determinants of SLT use

Determinants of SLT use referred to the reason and factors that influence firefighters for using SLT. Two categories emerged in this theme: (1) personal; and (2) social-environmental determinants, and were reflected across the majority of participant responses.

**Personal determinants.** *Positive perceptions on SLT*. Participants noted a number of positive attitudes toward using SLT. Safety and convenience of SLT use were cited as primary

**Table 2. Determinants of SLT use, barriers to cessation, and motivators for SLT cessation among participants.**

| Categories | Themes and subthemes |
|---|---|
| **Determinants of SLT use** | *Personal determinants* |
| | • Positive perceptions on SLT |
| | *Social-environmental determinants* |
| | • Influences from friends, family members and SLT users |
| | • Acceptability of SLT use in fire service |
| | • Coping with job stress |
| **Barriers to cessation** | *Intrapersonal barriers* |
| | • SLT use habits and dependency |
| | • Concerns about withdrawal symptoms |
| | *Social-environmental barriers* |
| | • Lack of support from health and other service providers |
| | • Lack of enforcement of existing tobacco policies |
| **Motivators for cessation** | *Internal motivators* |
| | • Self-motivation/self-decision |
| | • Health concerns |
| | *External motivators* |
| | • Social support from friends and family |
| | • Incentives |
| | • Price issues |

reasons for choosing SLT over cigarettes. Firefighters believed that SLT products are safer and less harmful than cigarettes, citing smoke inhalation or combustion and the range of health problems (e.g., lung function and capacity, cardiovascular health) associated with cigarette use as reasons.

*"I don't like smoking. I'm actually a runner. So that whole compromised to your respiratory capacity and volume. And you can feel it. It just feels incredibly different."* (P9, user)

*"I smoked prior, but knowing how much running we had to do, I quit that and sought the other avenue [chewing] because it was A, easier to do, but B, it didn't affect my running and things like that, so that was kind of the avenue I went with it. I've been doing it since then."* (P33, former)

Firefighters also noted that using SLT in their fire stations was discreet and convenient compared to smoking cigarettes. For some, the irritating and unpleasant cigarette smell made the participants decide to give up smoking and use SLT instead.

*"You don't stink. . .You don't have to worry about it being on clothes, making your car smell. It's clean. You spit in the [bottle], you throw the bottle away. There're no ashtrays. There's no cigarette butts. There's no, like I said, just the smell."* (P38, user)

*"You can't smoke with a mask on. But if you got a small enough dip, you can put a dip in, put your mask on, and go into a fire. It's not that big of a deal. It doesn't affect your mask that much. You can dip indoors when you're doing your paperwork or doing your training, or whatever the case may be. Smoking is all outdoors. You can't do any of that indoors."* (P24, former)

Firefighters referred to pleasure as another reason for chewing SLT. Both chewing and dipping SLT were frequently viewed as a means of killing time and calming down after fire duty responses. They also mentioned enjoying the feeling of happiness or relaxation after chewing SLT.

*". . .it relaxes me and lets me kind of take my mind off some stuff that's going on."* (P25, user)

*"I think a lot to me was some of the downtime because we work a 48/72 schedule here, so you're here for two days straight. Still in that timeframe, when it's down to when it's your own time, you're done working out and all that, for me it was just being bored, and this or that."* (P33, former)

**Social-environmental determinants.** *Influences from family members, friends, and SLT users.* Influences from family members, friends, and SLT users nearby were noted as the primary reason for the initiation and continuation of SLT use in our sample. In addition, similar stories about participants' first use experiences were noted. They often reported being introduced to and initiating SLT use through their family members and friends. Fathers, male cousins, and male friends were cited as the most important influences.

*"My family used it. I know my friends did it. I have an older brother, and all of his buddies did it. And of course, I tried it because I was the younger one and I had to try everything"* (P38, user)

*"I think probably the area we live in, it was just this really popular, smokeless tobacco. So I think at an early age, it was just always around. That's kind of what probably started me."* (P29, former)

Interestingly, late SLT initiation (i.e., starting use as an adult after joining the fire service) was reported by several participants. There was general agreement in this sub-group that they were introduced to SLT products by other firefighters.

*". . .regularly would be after joining the fire service. . .I don't think I was addicted to it until afterwards. And then it got to be a common thing to use every call"* (P20, user)

*"I did not use it before joining the fire service, and I was probably several years into the fire service before I even tried it. . .started off with the chewing tobacco and then went to snuff"* (P21, former)

*Acceptability of SLT use in the fire service.* Across all participants, there was a consistent notion that there were a high prevalence and social acceptability of SLT within fire service communities. Some participants noted that it was common for them to offer or receive other firefighters SLT as a sign of a close relationship or as being a team member or as a way to show "camaraderie". It is apparent that the use and the sharing of SLT are the actions that is socially acceptable and have become a norm within the fire community. Since the firefighting profession has a unique work and social environment, they heavily rely on their teamwork to perform their jobs involve not only fire suppression but providing emergency medical services, and rescuing operations. Thus, accepting or offering SLT is another way for firefighters to build trust, affirm social ties, and create a sense of camaraderie noted in the fire service.

*"We're in a dangerous situation and you've got those endorphins floating around and that team camaraderie. It's just like, Heck yeah. We did a great job. Hey, let's clean up. Hey, do you want a chew? Yeah. . .sure."* (P9, user)

The idea that SLT use is prevalent and widespread in the fire service was repeatedly mentioned. SLT use was perceived to be so common that seeing firefighters who chewed or dipped SLT in fire stations is normal.

*". . .I chew more at work because I see everybody else chewing. Which makes you think about doing it way more than if you're at home, and nobody else is chewing. . . But when you're at work, and you've got three or four other people that you live with using it all of the time, it makes you think about it a lot more."* (P13, user)

*"But like after regular hours. . . everybody can sit down in the kitchen or the TV room or even go to their own room and have a chew. And this goes on all throughout the day, though. When we're outside folding hose or doing hose evolutions or cleaning trucks, everybody's chewing or vaping or whatever. It's just one of those things"* (P38, user)

*Coping with job stress.* The use of SLT as a means to cope with stress was identified as another reason for using SLT. The nature of firefighting tasks and the stressful context of the job were frequently mentioned and presented as reasons why firefighters started and continued to use SLT. Those firefighting tasks include exposures to natural and human-caused disasters (e.g., rescuing victims, body recovery, fire extinguishing, and potential for injury and/or death). In addition, firefighters reported experiencing additional occupational stressors such

as sleep deprivation, erratic shift schedules, longer shift hours, and harassment [42, 43]. Several participants reported that using SLT helped with stress related to job and family responsibilities.

> *"There's a lot of stress that goes into it [jobs], and a lot of this and a lot of that. Sometimes people have to find comfort, and sometimes that's the comfort that they have is smokeless tobacco."* (P23, user)

> *"I don't want to blame it on stress, but we're in a very stressful-type job. . . always something going on. And just being able to do that helps me relax."* (P25, user)

## Barriers to cessation

Barriers were defined as factors that made SLT cessation difficult or prevented a firefighter from quitting SLT. The results reveal why they have difficulty quitting from the perspective of firefighters who were current and former SLT users. Two specific barriers to successful quitting emerged: (1) intrapersonal barriers (e.g., ingrained habit and dependency on SLT); (2) social-environmental barriers (e.g., lack of cessation resources in fire service).

**Intrapersonal barriers.** Intrapersonal barriers referred to the factors within the participant's self that interfered or impeded their quitting process such as habit of chewing/dipping SLT and nicotine dependency, and concerns about withdrawal symptoms.

*SLT use habits and dependency.* Firefighters stated that SLT was built into their daily lives, with participants acknowledging that chewing or dipping has become their habit and has become embedded in their daily routine. Others said it was difficult to break this habit and could not imagine their lives without SLT.

> *"Just that it's such a habit and such a part of my life. Everything reminds me of dipping, you know? I wake up, I want to put a dip in. I eat dinner, I eat breakfast or whatever, I got so I want to have a dip. I check out the truck so I want to put in a dip. Pretty much anything reminds me of that."* (P34, user)

> *"It got beyond. . .it was no longer an enjoyment thing. It more or less became a process of just, you know, if I was doing something, I just had it. I think the realization that it had become a habit, not a desire."* (P22, former)

SLT addiction or dependency was commonly identified as another barrier to quitting. SLT dependence was reinforced by the enjoyment and relaxation firefighters experienced from dipping or chewing tobacco. Use was also facilitated by the gratification firefighters get from chewing.

> *"It became an addiction to the point where I didn't need to chew, but I wanted to chew."* (P22, former)

*Concerns about withdrawal symptoms.* Firefighters reported overpowering withdrawal symptoms upon quitting, particularly when the symptoms interfered with their fire duty tasks or daily life. Participants expressed concerns about going through nicotine withdrawal, which they considered a complicated process. Some did not want to go through withdrawal and worried that withdrawal would distract them from performing firefighting tasks. Firefighting jobs also demand a high level of physical and mental ability to perform time pressure tasks and accurate decision making. Those who quit tobacco may experience various symptoms

including irritability, depressed mood, restlessness, increased appetite, and weight gain. Thus, it is likely that firefighters do not want to suffer through unpleasant feelings and discomforts.

> *"Probably the withdrawals as far as the headaches. I can deal with most of the other things, but usually a headache is—that doesn't go well with what I do for a living. . . And if you had a headache or any type of withdrawal, that takes away from your attentiveness to your job. And that's just [not] satisfactory."* (P20, user)

> *"They don't want to go through withdrawal. Or they want maybe a substitute that would help them ease off the tobacco. They don't want to go through the jitters or whatever goes along with it."* (P21, former)

**Social-environmental barriers.** Social-environmental barriers involved external factors that affected participants' involvement in the SLT cessation process, including lack of support from health and other service providers for SLT cessation and lack of enforcement of existing tobacco policies.

*Lack of support from health and other service providers.* Firefighters identified their views on the working environment (i.e., fire stations, fire departments) as obstructing their SLT quitting process. Lack of support to quit SLT from health and other service providers was noted. From what was described, it seemed to be relatively common for their Employee Assistance Program (EAP) to suggest SLT users use a smoking cessation program which they did not believe was useful or well-suited for SLT users who want to quit.

> *"Our health insurance that is provided does not have a tobacco cessation program in the EAP [Employee Assistance Program]. . . .So right now, you have to quit on your own, without any help or any programs. . .With our insurance and the EAP, we don't even have the smoking cessation program. We don't even have that option, let alone the smokeless tobacco cessation"* (P13, user)

> *"They don't really have anything on the market for smokeless. Everything is geared towards smokers. Getting a nicotine patch or something like that, you get the nicotine, but you don't get the feeling of it in your lip. And I think that is a big part of it."* (P32, user)

*Lack of enforcement of existing tobacco policies.* Participants also reported concerns about the limited SLT-specific cessation treatments, and a lack of restrictions on SLT use (or lack of enforcement of existing tobacco policies) in the departments as barriers. They were aware and familiar with the tobacco-free policies at their fire stations regarding to smoke-free policies; however, they were often unsure of specific rules and regulations on SLT use. They felt that policies should include or enforce SLT use.

> *"Even if the government or the department puts out a policy, no chewing tobacco or smoking within a station or only in designated areas, it's chewing tobacco. That policy really isn't enforced on chewing tobacco."* (P26, user)

## Motivators for cessation

Motivators were defined as factors that positively lead to behavioral change in SLT cessation and encourage participants to be SLT abstinence. Participants expressed and discussed their motivators for quitting, although not all firefighters desired to quit. Two categories emerged in this theme: (1) internal; and (2) external motivators.

**Internal motivators.**   Internal motivators refer to factors under a person's control that can motivate firefighters to quit using SLT. These involved self-motivation/self-decision and health reasons.

*Self-motivation/self-decision.* Firefighters primarily mentioned self-motivation and will-power as factors related to them not quitting or successfully quitting. They also described a belief that the success of quitting SLT lies within the power and mental preparedness of the individual that prompt them to be ready and commit to behavior change.

> *"It was literally just fighting the urge, you're letting your body do the withdrawal part, and retraining your bad habits. And everyone I know who's done it that way, including myself, has been successful."* (P22, former)

> *"There wasn't anything medical wrong. There weren't any dental problems or anything like that. It was just a self-decision that I need to quit. And each time it's been like that. Like. . .Ok, I just need to quit."* (P24, former)

*Health concerns.* Health concerns also were reported as motivators to quitting. Participants indicated that they were willing to stop using SLT to improve their overall health. The fear of health loss in the future was also mentioned.

> *"Now that I have children, a wife, and people that depend on me, it kind of makes me want to be around a lot longer and live a more healthy lifestyle and do whatever I can."* (P26, user)

> *"And then, currently, it [using SLT] weighs on me more and more just because of health bene-fits and risks and stuff like that. The older I get, the more and more it weighs on me really heavy to stop."* (P9, user)

**External motivators.**   External motivators refer to social and environmental factors beyond the persons' control which included family influences, incentives, and the cost of SLT products.

*Support from family and friends.* The majority of firefighters interviewed spoke of the importance of family members and friends being supportive during their SLT cessation pro-cess. Former users said family members encourage them to quit, played a crucial role in help-ing them through the withdrawal process, and provided moral support to go through the cessation process.

> *"When I quit last time, I had my wife hold me accountable, because I know she wasn't going to let me slip up. I don't think anything can replace that."* (P32, former)

> *". . . I made a promise to my wife that, you know. Okay, I was done. And we had two young boys. I had a one-year old and a three-year old. So that was a good time to quit back then."* (P35, former)

*Incentives.* Firefighters discussed the possibility of using rewards or incentives for quitting. Most participants brought up this idea and felt that incentives could influence an individual's motivation to quit. Types of incentives and rewards that might be effective included vouchers exchangeable for goods, offering a wellness/fitness program (i.e., gym membership), health premium insurance deduction, salary bonuses, and vacation days). Although some firefighters mentioned that receiving bonuses could promote quitting, most participants found that get-ting benefits from the departments such as paid time off, gym membership discounts, and health plan premium discounts may have a better effect.

*"Yes, a reward system, absolutely. They get an extra vacation day, or they get a small bonus like that to reward them for being tobacco free, which is going to help everybody. It's going to help them. It's going to help the department because it's going to decrease the chance of cancer."* (P13, user)

*"So as an incentive to me, "Oh hey, you're a non-smoker, non-tobacco user, we're going to give you a little bit of a cut on your health insurance premiums." So, that type of incentive means something to me. I think it gives incentive for the people that are using it also to be able to think, "Hey, I can lower my insurance premiums by whatever—10 percent for example." That might be an incentive to quit."* (P35, former)

*The cost of SLT products.* The cost of SLT use and the prospect of future price increases also were reported as motivators for quitting among participants. In response to the higher cost of SLT, firefighters described cutting down the number of daily use and eventually quitting SLT completely. The awareness of the price and the calculations of their monthly or yearly SLT spending were often said to have led to quit SLT.

*"Normally a can of Copenhagen or stuff can cost 6 bucks a can, and you're doing three or four cans a week times 52 weeks. . .it would actually save me money in the long run not buying it. So that's pretty much my incentive."* (P23, user)

*". . .the money piece, it's definitely become—it used to cost two dollars for a can, now it costs seven dollars for a can. And when you do the mathematics of it, you're saving a whole lot of money by not buying it. . . when I did the math on it, I figure in the last year of not chewing, I saved about 2500 dollars."* (P22, former)

## Discussion

To our knowledge, this is the first study to analyze personal, social, and environmental determinants related to SLT use in fire service, allowing for a greater understanding of factors contributing to SLT use among firefighters who were current SLT users and who successfully quit. As described previously, rates of firefighters' cigarette smoking have declined while SLT use rates have been increasing [13, 29]. Among male career firefighters, SLT use rates were three times higher than those found in the general male population [11]. Alarmingly, the fire service has comparable rates to other occupational groups with the highest rates of use [15, 16]. Despite the remarkably high prevalence of SLT use among firefighters, no non-tailored or occupationally -tailored SLT education or treatment programs exist for the fire service. Studies have identified factors that influence tobacco use in the general population [44] or even in the firehouse [14]; however, there is a dearth of qualitative research exploring possible reasons for firefighters' SLT use, as well as the barriers and motivators related to SLT cessation in this population.

Overall, positive perceptions on SLT (i.e., low health risk, convenience, discreet, relaxation), influences from SLT-using peers and family members, acceptability of SLT use in fire service, and using SLT to manage job stress were cited as the determinants for SLT use. The current findings support prior research indicating that perceptions of risks and benefits are predictive of continued tobacco use [45, 46]. Although participants were aware of health risks associated with SLT, they frequently framed SLT health risks in comparison to cigarette smoking, emphasizing that SLT is not as harmful as cigarettes. These similar perceptions on risks relative to cigarettes have been reported in other studies [32, 44, 47]. However, contradictory to participants' beliefs about the "safety" of SLT, it has been documented that SLT products

contained more than 30 carcinogens, including aldehydes and nitrosamines [48, 49]. In addition, SLT users were found to have higher NNK biomarkers levels, a tobacco-specific lung carcinogen, compared to cigarette smokers [50]. Understanding firefighters' perceptions regarding health and addiction risks are crucial to inform fire service policymakers and health promotion personnel because there clearly is a need to rectify misconceptions about SLT products.

Firefighters in our sample frequently mentioned beneficial perceptions of SLT use and workplace (or firehouses) acceptability in SLT. Several reasons were cited as possibly linked to the positive view of SLT products compared to cigarette smoking among firefighters [14]. For example, because the fire service is the primary responder to medical emergencies in the US, firefighters often witness the results of cigarette use on the populations they serve (e.g., fires started by cigarettes, smoking-related illness). Thus, it is not surprising that the national fire service has strongly voiced opposition to cigarette smoking and the hazards posed by smoking to lives and property. These same risks do not exist for SLT.

The present findings also highlight that family and peer influences can act as both a determinant for SLT use and a barrier to cessation. The influences from friends and family and other firefighters who are SLT users were found to be key factors in SLT initiation. While social influence on SLT initiation and continuation is not unique to the firefighters, it is certainly amplified by the high prevalence of SLT use within the fire service. Our study participants also frequently cited that being around other SLT-using firefighters makes their quit attempts challenging. A lack of support for quitting SLT from health professionals and fire service leadership also was identified as one of the main barriers. Evidence has demonstrated that brief counseling can increase the likelihood of successful quitting in SLT users [33]. Efforts should be focused on improving fire service health professionals or EAP's ability to offer quit advice sensitive to the fire service culture. In addition, concerns about the limited availability of SLT-specific cessation treatments and restrictions on SLT use (or lack of enforcement of existing tobacco policies) in the departments have been raised. This reflects the organizational level that neglects to address the unique issues of SLT that firefighters face.

Firefighters were primarily motivated to quit by the support system offered by their families. Both current and former SLT user groups indicated they would stop or had stopped using SLT due to their spouse. Several studies have shown that spouses' support is highly predictive of successful tobacco and SLT cessation [51–53]. Such support included cooperative behaviors (i.e., talking the users out of using tobacco) and reinforcement (i.e., expressing pleasure at the users' efforts to quit), which can encourage SLT quitting, help to buffer the stress from nicotine withdrawal process and the cue to use SLT. Another strong motivator for SLT cessation was incentives and rewards. Research shows that incentives, such as a gym membership or an extra paid day off, have increased participation in a wellness program and long-term quit rates [54]. More frequently, the price of SLT was another reason former SLT users gave for quitting, and current users considered quitting. This finding indicates that an increase in SLT price can influence motivation for quitting chewing or dipping SLT. Several states have raised taxes on SLT products to reduce SLT initiation in adolescents and to promote SLT cessation in adults [55]. State excise taxes effectively lower the use and frequency of SLT in both youths and adults [56, 57]. Numerous research studies have reported that raising SLT prices through taxation reduced SLT use rates and motivated users to quit SLT [56, 58, 59].

This study has a number of notable strengths. First, the current study provides a unique perspective on the factors influencing SLT use and barriers and motivators for SLT cessation in the fire service not easily captured in quantitative data. Second, data analysis procedures included multiple raters using an iterative and recursive process to best capture the themes of the data. Finally, the data provide one of the first glimpses into a professional or occupational

group that has successfully reduce smoking prevalence in a relatively short time, yet the rates of SLT use are high and rapidly increasing.

Despite the study's strengths, limitations to the findings exist. For instance, like most formative research, the sample size was limited in order to collect in-depth interview data. Regardless of the small sample size, there was strong convergence among participants regarding their opinions about SLT and SLT use in the fire service community. Although saturation was achieved, there may be selection biases in that more fire chiefs were presented in the interviews. Participant selection involved a purposive sampling of career firefighters and current- and former SLT users [30]. In this study, the sample being considered was limited to only career firefighters since the focus of this study was on opinions and thoughts that related to firefighters' occupational exposures. Thus, the perceptions and views regarding SLT captured may be more reflective of this sub-population than of SLT users in general populations or volunteer firefighters. Future studies should examine factors and perceptions/opinions related to SLT use in volunteer firefighters to determine whether the findings of this study generalize to other firefighting personnel.

The findings from the present study have further implications for cessation interventions targeting SLT-using firefighters. This study suggests that using a social cognitive theory-based approach to SLT cessation with this occupational group may be effective if it incorporates personal, social, and environmental contexts that commonly impact firefighters. Key findings suggest that interventions targeted to career firefighters should include specific modules on stress management techniques that are unique to this occupational group and educational materials aimed at changing culture or the fire service community's acceptability of SLT. The cessation programs should also consider the involvement of social support networks (e.g., friends and family, or firefighters who have stopped using SLT successfully) to provide support for those trying to quit. Social support might influence quitting through several ways, including influencing motivation to initiate quitting or encouraging to continue to abstain when facing withdrawal symptoms [60]. The benefits of the study also provide a foundation for future research, prevention, and SLT intervention efforts among fire service and research communities.

## Supporting information

**S1 File. Former and current SLT users interview guide with consent form.**
(PDF)

**S2 File. Consolidated criteria for reporting qualitative research (COREQ) checklist.**
(DOCX)

## Acknowledgments

The authors would like to thank the firefighters for their support and participation in the study.

## Author Contributions

**Conceptualization:** Nattinee Jitnarin, Sara A. Jahnke.

**Data curation:** Nattinee Jitnarin.

**Formal analysis:** Nattinee Jitnarin, Hannah N. Kelley.

**Funding acquisition:** Nattinee Jitnarin.

**Investigation:** Nattinee Jitnarin, Walker S. C. Poston.

**Methodology:** Nattinee Jitnarin, Walker S. C. Poston, Sara A. Jahnke.

**Project administration:** Hannah N. Kelley.

**Supervision:** Walker S. C. Poston, Sara A. Jahnke, Herbert H. Severson.

**Validation:** Walker S. C. Poston, Sara A. Jahnke, Christopher K. Haddock, Herbert H. Severson.

**Writing – original draft:** Nattinee Jitnarin.

**Writing – review & editing:** Nattinee Jitnarin, Walker S. C. Poston, Sara A. Jahnke, Christopher K. Haddock.

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
