## [Decision Letter · Decision Letter 0]

28 Jul 2020

PONE-D-20-18503

A qualitative study to assess perceptions, barriers, and motivators supporting smokeless tobacco cessation in the US fire service

PLOS ONE

Dear Dr. Jitnarin,

Thank you for submitting your manuscript to PLOS ONE. After careful consideration, we feel that it has merit but does not fully meet PLOS ONE’s publication criteria as it currently stands. Therefore, we invite you to submit a revised version of the manuscript that addresses the points raised during the review process.

In revising the paper, please pay special attention to the methodological issues raised by Reviewer 2's comments [2] and [4].  Per PLOS ONE's editorial policies, novelty is not an issue in the decision of whether or not to accept the paper, although the paper would be strengthened by making what is new in the paper clearer.  You should also address Reviewer 1's suggestions.

We look forward to receiving your revised manuscript.

Kind regards,

Stanton A. Glantz

Academic Editor

PLOS ONE

Journal Requirements:

2. Please note that according to our submission guidelines (http://journals.plos.org/plosone/s/submission-guidelines), outmoded terms and potentially stigmatizing labels should be changed to more current, acceptable terminology. For example: “Caucasian” should be changed to “white” or “of [Western] European descent” (as appropriate).

Reviewers' comments:

Reviewer's Responses to Questions

**Comments to the Author**

1. Is the manuscript technically sound, and do the data support the conclusions?

Reviewer #1: No

Reviewer #2: Yes

2. Has the statistical analysis been performed appropriately and rigorously? 

Reviewer #1: N/A

Reviewer #2: Yes

3. Have the authors made all data underlying the findings in their manuscript fully available?

Reviewer #1: No

Reviewer #2: Yes

4. Is the manuscript presented in an intelligible fashion and written in standard English?

Reviewer #1: Yes

Reviewer #2: Yes

5. Review Comments to the Author

Reviewer #1: This paper reports on findings from a qualitative study that explored firefighters’ experiences and perceptions of smokeless tobacco (SLT) use, as well as their motivations and barriers to SLT cessation. A strength of the study is that it is the first qualitative study to specifically explore firefighters’ experiences with SLT use and cessation, and firefighters are a group with particularly high rates of SLT use. I appreciate how much time goes into qualitative research and commend the authors for their hard work. There are, however, several major drawbacks in the current version of this paper, which I have outlined with suggestions for revision in the attached comments.

Reviewer #2: This is well designed and well-presented qualitative study on the use of smokeless tobacco among career firefighters in the US. It is an important topic as there is now strong evidence that firefighters in the US and elsewhere are at increased risk for some cancer sites, and there is an emerging body of literature on elucidating possible sources of that risk as well as prevention and control strategies that address firefighting culture. There are minor weaknesses to be addressed.

Introduction

• Starting on line 67: The Union is generally focused on Reducing Tobacco Use - not just smoking cessation, please revise the text to reflect this.

• The background on US workers rate of SLT use is missing some relevant information, specifically, (starting on line 73) “For example, we found that between 10.5%-18.4% of male firefighters exclusively used SLT which is almost three times the rate of SLT use found in the general population [13] and higher than any occupational group reported for the U.S. civilian workforce [17].” Citation 17 is Dietz et al., 2011 who present findings on four aggregated sectors of the US workers. However, three more recent studies that present findings from analysis of BRFSS, NSDUH, and NHIS data, each explore more refined worker sectors. All three of these studies reported that SLT use among extraction workers was similar, or higher, then the 10.5%-18.4% rate SLT among firefighters (please see citations below). This has important implications in terms of possible relevance of this paper to the extraction sector, particularly the recommendation for SLT cessation-treatment be tailored to the particular culture of firefighters (the authors do an excellent job of addressing this point) could be extended as a more general statement for other workforces.

Citations about SLT prevalence the extraction industry:

• Kristin Yeoman K, Sussell A, Retzer K , Poplin G. Health Risk Factors Among Miners, Oil and Gas Extraction Workers, Other Manual Labor Workers, and Nonmanual Labor Workers, BRFSS 2013-2017, 32 States. Workplace Health Saf. 2020 Aug;68(8):391-401. doi: 10.1177/2165079920909136. Epub 2020 Jun 2.

• Graber JM, Delnevo CD, Manderski MT, et al. Cigarettes, Smokeless Tobacco, and Poly-Tobacco Among Workers in Three Dusty Industries. J Occup Environ Med. 2016 May;58(5):477-84. doi: 10.1097/JOM.0000000000000699. PMID: 27158955

• Mazurek JM, Syamlal G, King BA, Castellan RM, Smokeless tobacco use among working adults - United States, 2005 and 2010. MMWR Morb Mortal Wkly Rep. 2014 Jun 6;63(22):477-82. PMID: 24898164 PMCID: PMC5779362

Methods

• Overall a very nice presentation and an unusually good balance with the amount and choice of quotes to support/demonstrate the themes. A few suggested changes:

o the 3 quotes for the ‘Social and community networks’ subheading (page 20, starting line 272) – this seems like a lot and the first quote seems to describe stress relief better than connectivity.

o on page 21, line 293 The quote could be shortened without impacting the content by deleting “The EAP does not have…a tobacco cessation program.” And replacing with “…”

o For the quote string on Page 23, line 336, consider defining “it” parenthetically with something like “[using SLT]”

• Was dual use of SLT and combustible tobacco explored? While smoking is bit lower among firefighters than the general public (cited here 13.2% vs 15.6%), dual tobacco use in na occuatanl setting is an important topic since smoking is much more effective nicotine-delivery methods and people who chew at work may smoke off shift. This may not be given as a reason for chewing at work by career firefighters without prompting given that there has been cultural shift away from combustible tobacco use

• The prevalence of SLT use varies significantly by region in the US, with highest rates in the South and West. This was national study with representation in 3 regions - (East, Central, and West). Were the authors able to see if there were differences on the cultural acceptance of SLT use within the fire service by region?

Results/Tables

• Please include the reporting for Table 1 in the results section

• Table 2 seems like the information does not need to be in a table but could go in the text as bullet points

• Table 3 seems more like a figure than a table?

• Line 130: please briefly describe the transcription QA p process

• Table 3, under barrier to cessation please is the lack of restriction for SLT use is in contrast to restrictions on combustible tobacco use, if so please make this explicit.

• Line 22: ‘personal barriers’ are called ‘Intrapersonal barriers’ on line 224, please make these terms consistent

Discussion

The concluding remarks include (staring line 456) “This approach would be particularly useful for fire service, where traditions and culture can be leveraged to enhance intervention acceptance and effectiveness.” is a central point, but please introduce this concept earlier in the discussion (page 20?) For the authors consideration: Might another possible intervention approach that capitalizes on fire house culture would be to engage the firefighters who are former SLT users as a support system for current users trying to quit?

The limitation of this study that it was conducted among career firefighters requires more explanation. Career firefighters make up about 33% of the US fire service and the vast majority work in urban population centers. In contrast, volunteer firefighters (67% of the US fire service) tend to serve in smaller rural communities where the prevalence if SLT is higher. As such there is a significant research gap that should be identified. As well, in key places in the manuscript, the authors should remind the readers that these are career firefighters, this should include (but not be limited to) lines: 65, 451, 82.

Other edits

• Line 86: please replace the word ‘using’ in the phrase “…a sample of current and former SLT using firefighters’ with the word ‘among’.

• Line 140: this is the first instance of the use of the acronym SCT, so please define it here

• Page 351, delete “by firefighters” (it is repetitive)

• Line 353, consider replacing “holidays” with “vacation days” as (in the US) generally ‘holiday’ implies a day off for everyone and vacation day implies an individual reward

6. PLOS authors have the option to publish the peer review history of their article (what does this mean?). If published, this will include your full peer review and any attached files.

Reviewer #1: No

Reviewer #2: **Yes: **Judith M Graber

---

## [Author Response · Author response to Decision Letter 0]

2 Sep 2020

We would like to thank the reviewers for their high quality and constructive reviews of our manuscript. We have revised and modified our manuscript per the reviewers’ comments and suggestions. Detailed point-by-point responses to each of the reviewers’ comments are follow: 

Reviewer #1: This paper reports on findings from a qualitative study that explored firefighters’ experiences and perceptions of smokeless tobacco (SLT) use, as well as their motivations and barriers to SLT cessation. A strength of the study is that it is the first qualitative study to specifically explore firefighters’ experiences with SLT use and cessation, and firefighters are a group with particularly high rates of SLT use. I appreciate how much time goes into qualitative research and commend the authors for their hard work. There are, however, several major drawbacks in the current version of this paper, which I have outlined with suggestions for revision in the attached comments.

We copied the reviewer#1 suggestions from the attached file and responded to their comments below. We thank the reviewer for their suggestions and now substantially edited and adjusted our manuscript per their comments. 

This paper reports on findings from a qualitative study that explored firefighters’ experiences and perceptions of smokeless tobacco (SLT) use, as well as their motivations and barriers to SLT cessation. A strength of the study is that it is the first qualitative study to specifically explore firefighters’ experiences with SLT use and cessation, and firefighters are a group with particularly high rates of SLT use. I appreciate how much time goes into qualitative research and commend the authors for their hard work. There are, however, several major drawbacks in the current version of this paper, which I have outlined below with suggestions for revision.

(1) The worthiness of the topic is not well established. 

The current introduction does not contextualize the research problem within the relevant existing literature, so the reader cannot be convinced that this is a problem worthy of study or imagine the practical or theoretical impact of the reported findings. The reader is left with many questions: 

What are the negative effects of smokeless tobacco use among fire fighters? In other words, why should we care if rates of use are high within this population? The introduction currently reviews the health effects of combustible tobacco.

We now provided additional details on the negative effects of SLT on firefighters. 

Have non-tailored SLT cessation interventions been implemented among firefighters in the past? If so, how successful were they? If none were implemented, what reasons can the authors provide to support their statement that tailored SLT interventions are necessary for this group rather than non-tailored interventions that already exist? In other words, if the justification for this study is that it can inform the development of tailored interventions for this group, a stronger case for the need for tailored interventions needs to be made in the introduction.

We provided additional explanations to support the statements that tailored SLT cessation programs for firefighters are necessary and needed to be implemented. 

Relatedly, why should we think that fire fighters would have unique perceptions/experiences regarding SLT as compared to other groups like construction workers or emergency medical technicians/paramedics? Why would we think that fire fighters’ perceptions/experiences with SLT would be different from their perceptions/experiences concerning combustible tobacco?

A number of reasons were considered for the unique cultures in fire service that may influence firefighters’ opinions and perspectives on SLT use including their long work hours (24-hour shifts or longer), the social and work bonding that occurs around their on-duty and off-duty days, and the stress associated with firefighting tasks. Although other occupations the reviewer mentioned have shift working structure similar to firefighters, they do not spend time (both live and work) at the same place in long hours like firefighters do in their fire station. Further, the significantly higher rate of SLT use compared to these other occupational groups and the general population rates suggest something specific to the fire service that is promoting use. 

Results from our previous study on tobacco use in a fire service personnel (Poston et al., 2012) showed that firefighters viewed and/or perceived SLT use and cigarette use (combustible tobacco) differently. For example, many firefighters noted that smoking negatively impacts their health and fitness and less acceptable among peer compared to SLT. Our findings from the current study also supported that SLT has been viewed positively and is viewed as acceptable in fire house more than cigarette smoking. 

I think answering these questions in the introduction will help to narrow and define the research problem and establish it as a more worthy and significant topic.

We revised the introduction section to address the reviewer’s concerns. 

(2) As it is currently presented, there is a lack of methodological coherence in how the study was designed, how the interview guide was developed, and the analytic strategy in deriving themes from the data. 

The sampling strategy is not well justified. For example, why include both current and former users in the same analysis? Wouldn’t a former user who has successfully quit potentially have different motivators and barriers to quitting than someone who has been using SLT for many years, tried to quit many times, and not been successful in doing so?

Data were analyzed separately for each group (current vs. former SLT users). We found similar themes across groups thus we presented results derived from both groups under the same themes. We added a sentence to clarify that the transcripts from current and former SLT users were coded and analyzed separately. 

Why was the interview guide developed with Social Cognitive Theory? The first mention of SCT comes a bit out of the blue in the methods section. No explanation of what SCT is or justification for its application to the interview guide is given. There is also mention of ‘previous literature’ that informed how the interview guide was developed. What findings/conceptual frameworks were taken from previous literature to help develop the interview guide? 

We have added the rationale/justification of using SCT in the introduction section and provided more details on using SCT for developing the interview guides in the method section. Findings from previous literature that were used to help develop the interview guide also are provided. 

The paper states that interviews ranged from 20 to 50 minutes. It does not seem credible that a meaningful discussion concerning all of the domains listed in the interview guide could be had in 20 minutes. This casts doubt on the quality of the data.

In one interview, we had to stop the interview early (20 minutes interview) due to participant’s responding to a fire call. To avoid the misunderstanding and resulting confusion, we now only report the average length of the interview.

As a reader, I experienced confusion regarding the analytic plan and how themes were derived from the data. For example, it appears that the transcripts were already partially coded and a coding scheme was developed prior to the application of a ‘SCT domain analysis and grounded theory’ approach. My understanding is that a methodological orientation, like grounded theory, should guide the methodological approach from the very beginning, rather than be introduced in the middle of the analytic process.

We revised the analytic plan and now added sentences to introduce the methodological approach at the beginning of data analysis section.

The paper states that “a thematic analysis by searching for patterns and themes that frequently occurred in a single interview or were common across interviews” was conducted. These sound like different analytic strategies. What was the rationale behind this?

In this step, we aimed to explore themes or patterns that frequently noted or occurred across interviews or groups. For example, under the existing SLT cessation services/supports in the fire service domain, both current and former SLT users frequently mentioned lack of support from their healthcare providers and from their fire departments regarding SLT cessation programs. 

If Social Cognitive Theory and ‘previous literature’ were used to develop the interview guide, I would expect these to also appear in the analytic strategy. Were some themes identified in advance to guide coding the transcripts (and if so these should be made explicit), or were themes derived entirely from the data?

We added an explanation on the use of SCT and how previous literature was used to develop the interview guide. We also added context about the domains that identified in advance to guide coding the transcripts. 

(3) The findings regarding participants’ experiences with and perceptions of SLT product use did not strike this reader as particularly novel. 

These included participants’ perceptions of reduced health risk with SLT use, the discretion and convenience of SLT products, and the use of SLT to cope with boredom and stress on the job. See, for example, young adults’ reported perceptions and experiences with smokeless tobacco in: McQuoid J, Keamy-Minor E, Ling PM. A practice theory approach to understanding poly-tobacco use. Critical Public Health. 2018 Nov 01. 10.1080/09581596.2018.1541226. 

Although some of the findings were similar to those found in the general population, we believe that the current study addresses a key gap in the occupational health literature with a group whose SLT rates are substantially higher than the US general population. We also believe the findings from this research will form the basis of developing an effective intervention to assist firefighters quit using SLT. 

The findings regarding the specific motivators for and barriers to quitting SLT for firefighters struck me as more of a contribution to the literature and seemed relevant to the development of a tailored SLT cessation intervention for fire fighters. I would emphasize these findings more in the next version of the paper. In particular: 

• Some firefighters worried that withdrawal would distract them from performing firefighting tasks

• The use of SLT to develop/strengthen camaraderie between firefighters

• Lack of support to quit SLT from health and other service providers. Concerns about the limited SLT specific cessation treatments, as well as a lack of restrictions on SLT use in the departments.

• Unique stressors of fire fighting

• Receptivity to an incentives program for quitting

• Sensitivity to SLT price increases

We revised our manuscript and emphasized the findings the reviewer’s raised in both results and discussion sections. 

(4) There is sometimes a lack of clarity and accuracy in how terms and constructs are used in the paper. 

In one example, the abstract refers to “the belief that it was effective for coping with boredom and stress”. This appears to be a statement of something that fire fighters have empirically experienced, so the term ‘belief’ does not seem appropriate. 

We have changed the term “belief” and revised the sentence in the abstract. We also edited the manuscript in line with the reviewer’s comments. 

Reviewer #2: This is well designed and well-presented qualitative study on the use of smokeless tobacco among career firefighters in the US. It is an important topic as there is now strong evidence that firefighters in the US and elsewhere are at increased risk for some cancer sites, and there is an emerging body of literature on elucidating possible sources of that risk as well as prevention and control strategies that address firefighting culture. There are minor weaknesses to be addressed.

We thank the reviewer for their suggestion and now address the weaknesses raised by the reviewer. 

Introduction

• Starting on line 67: The Union is generally focused on Reducing Tobacco Use - not just smoking cessation, please revise the text to reflect this.

This is revised in the current version. 

• The background on US workers rate of SLT use is missing some relevant information, specifically, (starting on line 73) “For example, we found that between 10.5%-18.4% of male firefighters exclusively used SLT which is almost three times the rate of SLT use found in the general population [13] and higher than any occupational group reported for the U.S. civilian workforce [17].” Citation 17 is Dietz et al., 2011 who present findings on four aggregated sectors of the US workers. However, three more recent studies that present findings from analysis of BRFSS, NSDUH, and NHIS data, each explore more refined worker sectors. All three of these studies reported that SLT use among extraction workers was similar, or higher, then the 10.5%-18.4% rate SLT among firefighters (please see citations below). This has important implications in terms of possible relevance of this paper to the extraction sector, particularly the recommendation for SLT cessation-treatment be tailored to the particular culture of firefighters (the authors do an excellent job of addressing this point) could be extended as a more general statement for other workforces.

Citations about SLT prevalence the extraction industry:

• Kristin Yeoman K, Sussell A, Retzer K , Poplin G. Health Risk Factors Among Miners, Oil and Gas Extraction Workers, Other Manual Labor Workers, and Nonmanual Labor Workers, BRFSS 2013-2017, 32 States. Workplace Health Saf. 2020 Aug;68(8):391-401. doi: 10.1177/2165079920909136. Epub 2020 Jun 2.

• Graber JM, Delnevo CD, Manderski MT, et al. Cigarettes, Smokeless Tobacco, and Poly-Tobacco Among Workers in Three Dusty Industries. J Occup Environ Med. 2016 May;58(5):477-84. doi: 10.1097/JOM.0000000000000699. PMID: 27158955

• Mazurek JM, Syamlal G, King BA, Castellan RM, Smokeless tobacco use among working adults - United States, 2005 and 2010. MMWR Morb Mortal Wkly Rep. 2014 Jun 6;63(22):477-82. PMID: 24898164 PMCID: PMC5779362

We revised the sentences and updated the citations to reflect the current findings on SLT prevalence in other occupational groups. 

Methods

• Overall a very nice presentation and an unusually good balance with the amount and choice of quotes to support/demonstrate the themes. A few suggested changes:

o the 3 quotes for the ‘Social and community networks’ subheading (page 20, starting line 272) – this seems like a lot and the first quote seems to describe stress relief better than connectivity.

o on page 21, line 293 The quote could be shortened without impacting the content by deleting “The EAP does not have…a tobacco cessation program.” And replacing with “…”

o For the quote string on Page 23, line 336, consider defining “it” parenthetically with something like “[using SLT]”

• Was dual use of SLT and combustible tobacco explored? While smoking is bit lower among firefighters than the general public (cited here 13.2% vs 15.6%), dual tobacco use in an occupational setting is an important topic since smoking is much more effective nicotine-delivery methods and people who chew at work may smoke off shift. This may not be given as a reason for chewing at work by career firefighters without prompting given that there has been cultural shift away from combustible tobacco use

We have edited the Methods section per the reviewer’s suggestions. We deleted the first quote under the “Social and community networks” subheading to make the section shorter. We also replaced “The EAP does not have…a tobacco cessation program.” with “…”, and added [using SLT] to refer to “it”.

• The prevalence of SLT use varies significantly by region in the US, with highest rates in the South and West. This was national study with representation in 3 regions - (East, Central, and West). Were the authors able to see if there were differences on the cultural acceptance of SLT use within the fire service by region?

Although SLT use in the general population varies significantly by region, our previous qualitative work in the fire service suggests that, although there are some regional differences for certain issues, there is a core common culture in the Fire Service, particularly in regards to substance use, particularly SLT [Poston WSC, Haddock CK, Jitnarin N, Jahnke SA. A national qualitative study of tobacco use among career firefighters. Nicotine Tob Res 2012;14:734-741].

Results/Tables

• Please include the reporting for Table 1 in the results section

We moved and reported the participants’ characteristics and Table 1 in the results section. 

• Table 2 seems like the information does not need to be in a table but could go in the text as bullet points

We eliminated Table 2 and now include the interview guide in S1 file. 

• Table 3 seems more like a figure than a table?

We reformatted Table 3 (now Table 2) so readers can easily follow the themes and subthemes relevant to the objectives of the study. 

• Line 130: please briefly describe the transcription QA process

We added a brief description for the transcription QA process. 

• Table 3, under barrier to cessation please is the lack of restriction for SLT use is in contrast to restrictions on combustible tobacco use, if so please make this explicit.

We clarified and changed “lack of restriction for SLT use” to “lack of enforcement of existing tobacco policies”. 

• Line 22: ‘personal barriers’ are called ‘Intrapersonal barriers’ on line 224, please make these terms consistent

We corrected the word “personal barriers” to “interpersonal barriers”. 

Discussion

The concluding remarks include (staring line 456) “This approach would be particularly useful for fire service, where traditions and culture can be leveraged to enhance intervention acceptance and effectiveness.” is a central point, but please introduce this concept earlier in the discussion (page 20?) For the authors consideration: Might another possible intervention approach that capitalizes on fire house culture would be to engage the firefighters who are former SLT users as a support system for current users trying to quit?

We have added sentences to introduce the concept in the beginning of the discussion. We also thank the reviewer for the suggestion on a support system. We are currently developing the intervention and using former SLT users as the support system is one of the components to assist firefighters to quit SLT. 

The limitation of this study that it was conducted among career firefighters requires more explanation. Career firefighters make up about 33% of the US fire service and the vast majority work in urban population centers. In contrast, volunteer firefighters (67% of the US fire service) tend to serve in smaller rural communities where the prevalence if SLT is higher. As such there is a significant research gap that should be identified. As well, in key places in the manuscript, the authors should remind the readers that these are career firefighters, this should include (but not be limited to) lines: 65, 451, 82.

We now emphasize that our study mainly focused on career firefighters per the reviewer’s suggestion and provided more explanations on career firefighters under the limitation of the study. 

Other edits

• Line 86: please replace the word ‘using’ in the phrase “…a sample of current and former SLT using firefighters’ with the word ‘among’.

• Line 140: this is the first instance of the use of the acronym SCT, so please define it here

• Page 351, delete “by firefighters” (it is repetitive)

• Line 353, consider replacing “holidays” with “vacation days” as (in the US) generally ‘holiday’ implies a day off for everyone and vacation day implies an individual reward

We edited our manuscript per the reviewer’s suggestions.

---

## [Decision Letter · Decision Letter 1]

11 Nov 2020

PONE-D-20-18503R1

A qualitative study to assess perceptions, barriers, and motivators supporting smokeless tobacco cessation in the US fire service

PLOS ONE

Dear Dr. Jitnarin,

Thank you for submitting your manuscript to PLOS ONE. After careful consideration, we feel that it has merit but does not fully meet PLOS ONE’s publication criteria as it currently stands. Therefore, we invite you to submit a revised version of the manuscript that addresses the points raised during the review process.

While you satisfied Reviewer 2, Reviewer 1 continued to recommend rejection of the paper, in even stronger terms than for the original submission.  For this reason I solicited a third review.  The third reviewer was also very critical and generally agreed with Reviewer 1, but thought you should be given a chance to try and fix the paper.  That is why I am giving you another chance to revise the paper.

The criticisms are, however, quite serious and will require a lot of work to fix.

Kind regards,

Stanton A. Glantz

Academic Editor

PLOS ONE

Reviewers' comments:

Reviewer's Responses to Questions

**Comments to the Author**

1. If the authors have adequately addressed your comments raised in a previous round of review and you feel that this manuscript is now acceptable for publication, you may indicate that here to bypass the “Comments to the Author” section, enter your conflict of interest statement in the “Confidential to Editor” section, and submit your "Accept" recommendation.

Reviewer #1: (No Response)

Reviewer #2: All comments have been addressed

Reviewer #3: (No Response)

2. Is the manuscript technically sound, and do the data support the conclusions?

Reviewer #1: No

Reviewer #2: Yes

Reviewer #3: Yes

3. Has the statistical analysis been performed appropriately and rigorously? 

Reviewer #1: N/A

Reviewer #2: Yes

Reviewer #3: N/A

4. Have the authors made all data underlying the findings in their manuscript fully available?

Reviewer #1: No

Reviewer #2: Yes

Reviewer #3: No

5. Is the manuscript presented in an intelligible fashion and written in standard English?

Reviewer #1: Yes

Reviewer #2: Yes

Reviewer #3: Yes

6. Review Comments to the Author

Reviewer #1: (No Response)

Reviewer #2: The authors have reviewed each of the comments and give very satisfactory responses. Many of my suggested reviews were editorial in nature and these were all addressed. My primary content concern was about variation in prevalence of smokeless tobacco use between the regions in the United States. The authors response that of smokeless tobacco use is a core common culture in the US fire service was based on their previous qualitative work in the fire service and was adequality addressed in their revised manuscript. I was also very concerned that the manuscript was not clear that they were focused on career (vs volunteer) firefighters and they have addressed this concern.

Reviewer #3: The revised manuscript addressed most, but not all of the concerns raised by the reviewers in the first round. There are also some additional issues that were not raised earlier, but need to be clarified.

The biggest issue with this paper is the inconsistency: inconsistency between the use of SCT to design the study and grounded theory approach to analyze the results; inconsistency in the use of terminology, and inconsistency between the SCT constructs and the reported results. Below, I first describe these structural inconsistencies and then detail other, more minor issues.

It seems that SCT was used to design the study (to develop the interview guide) and to guide the coding. This is fine. However, the authors specify that a grounded theory approach was used to analyze the transcripts. I believe that a grounded theory approach is incompatible with an a priori selection of a theory to guide the development of the study and the analysis of the interviews. The way I understand it, a grounded theory approach implies complete openness and allowing the data to dictate the emerging themes. This study, in contrast, was guided by the SCT from the start. (This issue was raised in the first round of reviews, and simply shifting the order of the presentation for methods does not resolve it.)

The described interview guide does not quite match the SCT structure. As the authors themselves described in the introduction, two of the main aspects of the SCT is observational learning and self-efficacy. The structure for “general domains” does not list those at all. This might stem from the use of different terminology (next point).

Throughout the paper, the terminology keeps changing, sometimes reflecting the SCT constructs (“personal, social, and environmental determinants” in the introduction) and at other times something else (“cultural, community, and personal influences” in the beginning of the discussion). In the results, the structure is completely different once again. It would be helpful to consistently use the same language to refer to the same issues.

To resolve these issues regarding theoretical inconsistency, I suggest the following approach (although other approaches might be useful): first, the SCT is described in the introduction to give the readers the overview of the theory. It should be described in such a way that would match the constructs that are used in the results. It could be mentioned that SCT was used as a loose theoretical guidance only in that it indicated the broad domains (intrapersonal, environmental, etc.). Second, the description on the data analysis in the methods should specify how SCT was used to guide the analysis and whether the themes were derived deductively or inductively. If grounded theory is mentioned, it needs to be clearly explained how it was used in a combination with an a priori theoretical framework of SCT. Throughout the paper it would be helpful to keep the structure consistent.

In the discussion, it would be worth addressing the following issues:

Factors influencing SLT use comprise two separate categories – factors that led to initiation of the SLT use (use for the first time) and factors that make firefighters use SLT on a regular basis. Furthermore, the barriers to quitting and factors influencing SLT seem to be very similar – do they work differently as barriers vs. facilitators? For example, “social influences” under “Factors influencing SLT use” are very similar to “social and community networks” under “barriers to cessation”. Why are they called differently when they are essentially the same?

Minor issues

It is unclear if all transcripts were coded by two coders and a subset (n=5) by three coders, or if all transcripts were coded by one coder and a subset (n=5) by two coders. If the latter, then it is unclear how the discrepancies were identified and reconciled in the transcripts that were coded only by one coder.

It should be specified explicitly somewhere that all participants were men (and why women were not included – because of low SLT rates among women).

Were firefighters who also smoked excluded? (I.e., were these exclusive SLT users or dual users of SLT and cigarettes?) Among former SLT users, were any using other tobacco products? It would be helpful to report this information.

Some of the information in the results would be more appropriate for the discussion. For example, information on raising taxes on SLT in the section on external motivators belongs in the discussion.

Proofreading would be helpful: “SLT cessation programs that tailored” , “traditional, teamwork, and mission can be leveraged”

In the results, stress is mentioned in multiple places and it would be better to put it in one place. For example, it might not be needed in the “organizational barriers”.

“Beneficial perceptions of SLT use, such as relaxation, focus, and stress-management, appeared to be more important barriers to promoting SLT cessation among our study participants than perceived disadvantage around health issues, money, and addiction, which have reportedly been strong motivators in tobacco cessation in previous work [50].” – It is unclear how the comparative importance was evaluated from the qualitative results; in the results, each of the categories is reported separately. In order to be able to conclude about the comparative importance, it is necessary to demonstrate that in the results first, but it is unclear if such analysis was made. The way the results are presented now can be described as “descriptive statistics” – they simply describe the themes and issues that participants mentioned, without more in-depth analysis of how they interacted and how they, in combination, might drive the behavior. It is fine to do a descriptive presentation, but it is important to make sure that the discussion is based on the results and does not speculate without the evidence.

From the first round of reviews, it does not seem like the following questions were sufficiently answered:

“Have non-tailored SLT cessation interventions been implemented among firefighters in the past? If so, how successful were they?” (The text states that culturally tailored interventions are nonexistent, but does not seem to address the issue of non-tailored SLT interventions).

Regarding reporting the length of the interviews – it would be more appropriate to report the range (with the explanation that one interview had to be cut short due to a response to a fire call) and the median length, rather than the average length of the interview.

And a final note on the format of responses to the reviewers’ comments. The reviewers provide detailed and thoughtful comments, and it is expected that the responses would be equally detailed. It is difficult for a reviewer to evaluate this response: “We now provided additional details on the negative effects of SLT on firefighters.” The reviewer now has to go through the article and try and figure out where these “additional details” are and if they are sufficient. It would be a lot easier for a reviewer to evaluate a response if it included a quote with that new information directly in the response letter. (It might lengthen the response letter significantly, but in the end it would save reviewers a lot of time and make them happier.)

7. PLOS authors have the option to publish the peer review history of their article (what does this mean?). If published, this will include your full peer review and any attached files.

Reviewer #1: No

Reviewer #2: No

Reviewer #3: No

---

## [Author Response · Author response to Decision Letter 1]

7 Mar 2021

Our responses to the editor and reviewers are in bold texts below:

6. Review Comments to the Author

Reviewer #1: (No Response)

Reviewer #2: The authors have reviewed each of the comments and give very satisfactory responses. Many of my suggested reviews were editorial in nature and these were all addressed. My primary content concern was about variation in prevalence of smokeless tobacco use between the regions in the United States. The authors response that of smokeless tobacco use is a core common culture in the US fire service was based on their previous qualitative work in the fire service and was adequality addressed in their revised manuscript. I was also very concerned that the manuscript was not clear that they were focused on career (vs volunteer) firefighters and they have addressed this concern.

We thank Reviewer #2 for their suggestions and comments, which helps to strengthen our manuscript. 

Reviewer #3: The revised manuscript addressed most, but not all of the concerns raised by the reviewers in the first round. There are also some additional issues that were not raised earlier, but need to be clarified.

We would like to thank the reviewer for their high quality and constructive reviews of our manuscript. We have revised and modified our manuscript per the reviewers’ comments and suggestions. Detailed point-by-point responses to each of the reviewers’ comments are as follow: 

The biggest issue with this paper is the inconsistency: inconsistency between the use of SCT to design the study and grounded theory approach to analyze the results; inconsistency in the use of terminology, and inconsistency between the SCT constructs and the reported results. Below, I first describe these structural inconsistencies and then detail other, more minor issues.

It seems that SCT was used to design the study (to develop the interview guide) and to guide the coding. This is fine. However, the authors specify that a grounded theory approach was used to analyze the transcripts. I believe that a grounded theory approach is incompatible with an a priori selection of a theory to guide the development of the study and the analysis of the interviews. The way I understand it, a grounded theory approach implies complete openness and allowing the data to dictate the emerging themes. This study, in contrast, was guided by the SCT from the start. (This issue was raised in the first round of reviews, and simply shifting the order of the presentation for methods does not resolve it.)

The described interview guide does not quite match the SCT structure. As the authors themselves described in the introduction, two of the main aspects of the SCT is observational learning and self-efficacy. The structure for “general domains” does not list those at all. This might stem from the use of different terminology (next point).

Throughout the paper, the terminology keeps changing, sometimes reflecting the SCT constructs (“personal, social, and environmental determinants” in the introduction) and at other times something else (“cultural, community, and personal influences” in the beginning of the discussion). In the results, the structure is completely different once again. It would be helpful to consistently use the same language to refer to the same issues.

To resolve these issues regarding theoretical inconsistency, I suggest the following approach (although other approaches might be useful): first, the SCT is described in the introduction to give the readers the overview of the theory. It should be described in such a way that would match the constructs that are used in the results. It could be mentioned that SCT was used as a loose theoretical guidance only in that it indicated the broad domains (intrapersonal, environmental, etc.). Second, the description on the data analysis in the methods should specify how SCT was used to guide the analysis and whether the themes were derived deductively or inductively. If grounded theory is mentioned, it needs to be clearly explained how it was used in a combination with an a priori theoretical framework of SCT. Throughout the paper it would be helpful to keep the structure consistent.

We now revised our manuscript based on the reviewer’s suggestive approach. First, we revised the introduction to match the SLT constructs that are used in the results (Line numbers: 102-116). Second, we revised our data analysis section (Line numbers: 161-182) to reflect on the approaches we used for data analysis and provided the description of how SCT was used to guide the analysis. Third, we now used the same language and terminology of SCT constructs throughout the manuscript. Lastly, we rearranged and presented our results according to themes and subthemes. 

In the discussion, it would be worth addressing the following issues:

Factors influencing SLT use comprise two separate categories – factors that led to initiation of the SLT use (use for the first time) and factors that make firefighters use SLT on a regular basis. Furthermore, the barriers to quitting and factors influencing SLT seem to be very similar – do they work differently as barriers vs. facilitators? For example, “social influences” under “Factors influencing SLT use” are very similar to “social and community networks” under “barriers to cessation”. Why are they called differently when they are essentially the same?

We provided the definitions of these two categories: determinants of SLT use: “Determinants of SLT use referred to the reason and factors that influence firefighters for using SLT” (Lines number: 199-201) and barriers to cessation: “Barriers were defined as factors that made SLT cessation difficult or prevented a firefighter from quitting SLT” (Line numbers: 292-293) to avoid confusion. 

We also arranged the findings to match our major themes per the reviewer’s suggestion. Now the subtheme “social and community networks” under “barrier to cessation” is renamed to “acceptability of SLT use in the fire service” and placed under “determinant of SLT use” (Line numbers 258-277).

Minor issues

It is unclear if all transcripts were coded by two coders and a subset (n=5) by three coders, or if all transcripts were coded by one coder and a subset (n=5) by two coders. If the latter, then it is unclear how the discrepancies were identified and reconciled in the transcripts that were coded only by one coder.

We now clarified and revised the sentence that two coders independently coded all 23 transcripts and the third researcher coded a subset of the transcripts (n = 5). 

The new sentence is “Two primary coders independently coded all 23 transcripts, compared their analyses, and discussed any discrepancies until consensus was reached. Coding was regularly reviewed and verified by the third researcher to identify and incorporate emergent themes and for data validity. In addition, the third researcher conducted independent coding of a subset of data (n = 5) to ensure that the final coding scheme had adequate reliability.” (Line numbers: 177-182).

It should be specified explicitly somewhere that all participants were men (and why women were not included – because of low SLT rates among women).

We added the sentence indicated that women firefighters were not included in the study due to the very low SLT use prevalence compared with men in the fire service: “In addition, women firefighters were not included in the study due to the very low SLT use prevalence compared with men in the fire service (0.5%; [31]).” (Line numbers: 133-135). 

We also reported in the result section that all participants were all men (Line number: 184).

Were firefighters who also smoked excluded? (I.e., were these exclusive SLT users or dual users of SLT and cigarettes?) Among former SLT users, were any using other tobacco products? It would be helpful to report this information.

We revised and clarified the definition of current and former SLT users in the method section. The revision sentence was changed to “current users - those who reported exclusively used SLT on multiple occasions within the past month; and former users - those who reported using SLT regularity in the past, but not currently using SLT or any tobacco products.” (Line numbers: 128-130). 

Some of the information in the results would be more appropriate for the discussion. For example, information on raising taxes on SLT in the section on external motivators belongs in the discussion.

This information on raising taxes on SLT was moved to the discussion section (Line numbers: 510-514).

Proofreading would be helpful: “SLT cessation programs that tailored”, “traditional, teamwork, and mission can be leveraged”

We proofread the whole manuscript and corrected the grammatical errors. 

In the results, stress is mentioned in multiple places and it would be better to put it in one place. For example, it might not be needed in the “organizational barriers”.

The stress subthemes were combined and moved under the “determinants of SLT use” and renamed to “coping with job stress” (Line numbers 278-290).

“Beneficial perceptions of SLT use, such as relaxation, focus, and stress-management, appeared to be more important barriers to promoting SLT cessation among our study participants than perceived disadvantage around health issues, money, and addiction, which have reportedly been strong motivators in tobacco cessation in previous work [50].” – It is unclear how the comparative importance was evaluated from the qualitative results; in the results, each of the categories is reported separately. In order to be able to conclude about the comparative importance, it is necessary to demonstrate that in the results first, but it is unclear if such analysis was made. The way the results are presented now can be described as “descriptive statistics” – they simply describe the themes and issues that participants mentioned, without more in-depth analysis of how they interacted and how they, in combination, might drive the behavior. It is fine to do a descriptive presentation, but it is important to make sure that the discussion is based on the results and does not speculate without the evidence.

This sentence is now removed from the manuscript since it is based on speculation, not the findings. 

From the first round of reviews, it does not seem like the following questions were sufficiently answered:

“Have non-tailored SLT cessation interventions been implemented among firefighters in the past? If so, how successful were they?” (The text states that culturally tailored interventions are nonexistent, but does not seem to address the issue of non-tailored SLT interventions).

Unfortunately, the non-tailored SLT cessation interventions have not been implemented in the fire service. We now addressed the issue of non-tailored SLT interventions in fire service by revising the sentence to “Additionally, no non-tailored of the SLT education or treatment programs has been implemented or provided for the fire service, and no occupationally tailored SLT cessation intervention exists for firefighters” (Line numbers: 93-95), and we mentioned it again in the discussion section: “…no non-tailored or occupationally-tailored SLT education or treatment programs exist for the fire service.” (Lines numbers: 445-447).

Regarding reporting the length of the interviews – it would be more appropriate to report the range (with the explanation that one interview had to be cut short due to a response to a fire call) and the median length, rather than the average length of the interview.

We now reported the range and median length of the interview to explain that one interview was only 20 minutes due to fire response: “Researchers (NJ, CP, HK) carried out the semi-structured interviews, ranging from 20 to 50 minutes (median 42 minutes). Of the 23 interviewed participants, one participant responded to a fire call, resulting in only 20 minutes of the interview.” (Line numbers: 157-159). 

And a final note on the format of responses to the reviewers’ comments. The reviewers provide detailed and thoughtful comments, and it is expected that the responses would be equally detailed. It is difficult for a reviewer to evaluate this response: “We now provided additional details on the negative effects of SLT on firefighters.” The reviewer now has to go through the article and try and figure out where these “additional details” are and if they are sufficient. It would be a lot easier for a reviewer to evaluate a response if it included a quote with that new information directly in the response letter. (It might lengthen the response letter significantly, but in the end it would save reviewers a lot of time and make them happier.)

We would like to thank the reviewer for the suggestion. We now included a quote with the new or additional information responding to reviewers’ comments. We also provided line numbers corresponding to the revised manuscript to navigate the reviewer to our responses.

---

## [Decision Letter · Decision Letter 2]

29 Mar 2021

PONE-D-20-18503R2

A qualitative study to assess perceptions, barriers, and motivators supporting smokeless tobacco cessation in the US fire service

PLOS ONE

Dear Dr. Jitnarin,

Thank you for submitting your manuscript to PLOS ONE. After careful consideration, we feel that it has merit but does not fully meet PLOS ONE’s publication criteria as it currently stands. Therefore, we invite you to submit a revised version of the manuscript that addresses the points raised during the review process.

We look forward to receiving your revised manuscript.

Kind regards,

Stanton A. Glantz

Academic Editor

PLOS ONE

Journal Requirements:

Reviewers' comments:

Reviewer's Responses to Questions

**Comments to the Author**

1. If the authors have adequately addressed your comments raised in a previous round of review and you feel that this manuscript is now acceptable for publication, you may indicate that here to bypass the “Comments to the Author” section, enter your conflict of interest statement in the “Confidential to Editor” section, and submit your "Accept" recommendation.

Reviewer #3: (No Response)

2. Is the manuscript technically sound, and do the data support the conclusions?

Reviewer #3: Yes

3. Has the statistical analysis been performed appropriately and rigorously? 

Reviewer #3: N/A

4. Have the authors made all data underlying the findings in their manuscript fully available?

Reviewer #3: Yes

5. Is the manuscript presented in an intelligible fashion and written in standard English?

Reviewer #3: Yes

6. Review Comments to the Author

Reviewer #3: The revisions mostly sufficiently addressed the issues raised in the previous round of review. The issue with the lack SCT constructs and the presentation of the results has been mostly resolved. There is one issue with the lack of clarity regarding the conceptual distinction between determinants and barriers, described below.

The section “Acceptability of SLT use in the fire service” is in the section on “Determinants of SLT use”, but it talks about barriers to cessation: “The high prevalence and social acceptability of SLT within fire service communities were cited as making quit attempts challenging. Some participants expressed that offering SLT or accepting SLT from other firefighters was the reason for their failure to quit, i.e., it was common for them to offer or receive other firefighters SLT as a sign of a close relationship or as being a team member or as a way to show “camaraderie”.” It would be better to rewrite these sentences to focus on social norms and widespread use as leading to use, rather than preventing the cessation.

The same issue is with the next section – “coping with job stress”: “The nature of firefighting tasks and the stressful context of the job were frequently mentioned and presented as unique barriers to SLT cessation for firefighters”. If it’s in the “Determinants of SLT use” section, it should be discussed as a determinant.

Related to the two examples above, it seems that in the previous version they were in the “barriers to cessation” section and then were simply moved from the “barriers” to “determinants” section. This goes back to my comment from the previous round – how are “determinants” and “barriers” conceptually different? Clearly, some things that make SLT use attractive in the first place make it harder to quit too (e.g., widespread use by peers). It might be worthwhile to indicate that some of the “determinants” were discussed as both “determinants” and “barriers” and that the section “barriers to cessation” explicitly deals with only those issues that focus on preventing cessation (or something like that).

In addition to this issue, there are some additional minor issues with language and focus, detailed next.

“Additionally, no non-tailored of the SLT education or treatment programs have been implemented or provided for the fire service, and no occupationally tailored SLT cessation intervention exists for firefighters.” – suggest changing to: “Additionally, no SLT education or treatment programs have been implemented for the fire service, and no occupationally tailored SLT cessation interventions exist for firefighters.”

“current users - those who reported exclusively used SLT on multiple occasions” – should be “current users - those who reported exclusively using SLT on multiple occasions”

Spell out “EAP” the first time it is used.

For Table 1, it might be more informative to provide the min and max of age years and time in service (years). Since there were two groups (current users and former users), it might be useful to provide separate columns for them. This way Table 1 will give a lot more information without repeating what is already said in the text.

This quote seems to belong in the section on “lack of support from health and other service providers” rather than in the section on “lack of enforcement of existing tobacco policies” where it currently is: “They don't really have anything on the market for smokeless. Everything is geared towards smokers. Getting a nicotine patch or something like that, you get the nicotine, but you don't get the feeling of it in your lip. And I think that is a big part of it.” (P32,

445 user)”

The paper needs to be carefully proofread to make sure the changes that were made correspond to the language left over from the earlier version. For example, the discussion lists 3 barriers for SLT cessation (“Given that this qualitative study identified three primary barriers for SLT cessation in fire service personnel”), but the section on barriers has either 2 or 4 (depending on the level).

Finally, in the discussion, there is a lot of general talk about how the study can inform the intervention and how a targeted intervention would be better. This information appears at least three times, and all those can be condensed. Recommendations for an intervention are likewise dispersed throughout the discussion. It would be better to clearly state – based on the findings, what an intervention (or series of interventions) for career firefighters should look like?

7. PLOS authors have the option to publish the peer review history of their article (what does this mean?). If published, this will include your full peer review and any attached files.

Reviewer #3: No

---

## [Author Response · Author response to Decision Letter 2]

18 Apr 2021

Our responses to the editor and reviewers are in bold texts below:

Thank you for submitting your manuscript to PLOS ONE. After careful consideration, we feel that it has merit but does not fully meet PLOS ONE’s publication criteria as it currently stands. Therefore, we invite you to submit a revised version of the manuscript that addresses the points raised during the review process.

We would like to thank the Academic Editor for the careful reading and the reviewers for their high quality and constructive review of our manuscript. We have revised and edited the manuscript in response to the extensive and insightful reviewers’ comments. We hope that a revised version of the manuscript will be considered by PLOS ONE.

We now include this rebuttal letter, the marked-up copy of our manuscript with track changes indicating edits made per the reviewers’ suggestions, and the unmarked version of our revised manuscript.

We look forward to receiving your revised manuscript.

Kind regards,

Stanton A. Glantz

Academic Editor

PLOS ONE

Journal Requirements:

Reviewers' comments:

Reviewer's Responses to Questions

Comments to the Author

1. If the authors have adequately addressed your comments raised in a previous round of review and you feel that this manuscript is now acceptable for publication, you may indicate that here to bypass the “Comments to the Author” section, enter your conflict of interest statement in the “Confidential to Editor” section, and submit your "Accept" recommendation.

Reviewer #3: (No Response)

2. Is the manuscript technically sound, and do the data support the conclusions?

Reviewer #3: Yes

3. Has the statistical analysis been performed appropriately and rigorously?

Reviewer #3: N/A

4. Have the authors made all data underlying the findings in their manuscript fully available?

Reviewer #3: Yes

5. Is the manuscript presented in an intelligible fashion and written in standard English?

Reviewer #3: Yes

6. Review Comments to the Author

Reviewer #3: The revisions mostly sufficiently addressed the issues raised in the previous round of review. The issue with the lack SCT constructs and the presentation of the results has been mostly resolved. There is one issue with the lack of clarity regarding the conceptual distinction between determinants and barriers, described below.

We would like to thank the reviewer for their high quality and constructive reviews of our manuscript. We have revised and modified our manuscript per the reviewers’ comments and suggestions. Detailed point-by-point responses to each of the reviewers’ comments are as follow: 

The section “Acceptability of SLT use in the fire service” is in the section on “Determinants of SLT use”, but it talks about barriers to cessation: “The high prevalence and social acceptability of SLT within fire service communities were cited as making quit attempts challenging. Some participants expressed that offering SLT or accepting SLT from other firefighters was the reason for their failure to quit, i.e., it was common for them to offer or receive other firefighters SLT as a sign of a close relationship or as being a team member or as a way to show “camaraderie”.” It would be better to rewrite these sentences to focus on social norms and widespread use as leading to use, rather than preventing the cessation.

We revised the “Acceptability of SLT use in the fire service” section to focus on social norms and widespread of SLT use. (Line numbers: 257-281).

The same issue is with the next section – “coping with job stress”: “The nature of firefighting tasks and the stressful context of the job were frequently mentioned and presented as unique barriers to SLT cessation for firefighters”. If it’s in the “Determinants of SLT use” section, it should be discussed as a determinant.

We also revised the “Coping with job stress” section to be more related to the determinants of SLT use. (Line numbers: 282-295).

Related to the two examples above, it seems that in the previous version they were in the “barriers to cessation” section and then were simply moved from the “barriers” to “determinants” section. This goes back to my comment from the previous round – how are “determinants” and “barriers” conceptually different? Clearly, some things that make SLT use attractive in the first place make it harder to quit too (e.g., widespread use by peers). It might be worthwhile to indicate that some of the “determinants” were discussed as both “determinants” and “barriers” and that the section “barriers to cessation” explicitly deals with only those issues that focus on preventing cessation (or something like that).

We thank the reviewer for their suggestions. We now revised the discussion section to reflect on our findings. 

In addition to this issue, there are some additional minor issues with language and focus, detailed next.

“Additionally, no non-tailored of the SLT education or treatment programs have been implemented or provided for the fire service, and no occupationally tailored SLT cessation intervention exists for firefighters.” – suggest changing to: “Additionally, no SLT education or treatment programs have been implemented for the fire service, and no occupationally tailored SLT cessation interventions exist for firefighters.”

We revised this sentence per the reviewer’s suggestion (Line numbers: 93-95).

“current users - those who reported exclusively used SLT on multiple occasions” – should be “current users - those who reported exclusively using SLT on multiple occasions”

We revised this sentence per the reviewer’s suggestion (Line numbers: 127-128).

Spell out “EAP” the first time it is used.

We now provided the full term of the EAP when it first mentioned (Line number: 349). 

For Table 1, it might be more informative to provide the min and max of age years and time in service (years). Since there were two groups (current users and former users), it might be useful to provide separate columns for them. This way Table 1 will give a lot more information without repeating what is already said in the text.

We edited Table 1 and provided the demographic characteristics for both former- and current users in separate columns. We also provided the range (min-max) of age (years) and time in service (years). 

This quote seems to belong in the section on “lack of support from health and other service providers” rather than in the section on “lack of enforcement of existing tobacco policies” where it currently is: “They don't really have anything on the market for smokeless. Everything is geared towards smokers. Getting a nicotine patch or something like that, you get the nicotine, but you don't get the feeling of it in your lip. And I think that is a big part of it.” (P32,

445 user)”

We moved this quote to the appropriate section, “lack of support from health and other service providers” (Line numbers: 357-360).

The paper needs to be carefully proofread to make sure the changes that were made correspond to the language left over from the earlier version. For example, the discussion lists 3 barriers for SLT cessation (“Given that this qualitative study identified three primary barriers for SLT cessation in fire service personnel”), but the section on barriers has either 2 or 4 (depending on the level).

We proofread the whole manuscript and corrected the grammatical errors. 

Finally, in the discussion, there is a lot of general talk about how the study can inform the intervention and how a targeted intervention would be better. This information appears at least three times, and all those can be condensed. Recommendations for an intervention are likewise dispersed throughout the discussion. It would be better to clearly state – based on the findings, what an intervention (or series of interventions) for career firefighters should look like?

We would like to thank the reviewer for the suggestion. We now revised the discussion section to be clearer and more succinct. 

7. PLOS authors have the option to publish the peer review history of their article (what does this mean?). If published, this will include your full peer review and any attached files.

Do you want your identity to be public for this peer review? For information about this choice, including consent withdrawal, please see our Privacy Policy.

Reviewer #3: No

---

## [Editor Report · Decision Letter 3]

21 Apr 2021

A qualitative study to assess perceptions, barriers, and motivators supporting smokeless tobacco cessation in the US fire service

PONE-D-20-18503R3

Dear Dr. Jitnarin,

We’re pleased to inform you that your manuscript has been judged scientifically suitable for publication and will be formally accepted for publication once it meets all outstanding technical requirements.

Kind regards,

Stanton A. Glantz

Academic Editor

PLOS ONE
---

## [Editor Report · Acceptance letter]

3 May 2021

PONE-D-20-18503R3 

A qualitative study to assess perceptions, barriers, and motivators supporting smokeless tobacco cessation in the US fire service 

Dear Dr. Jitnarin:

I'm pleased to inform you that your manuscript has been deemed suitable for publication in PLOS ONE. Congratulations! Your manuscript is now with our production department. 

Kind regards, 

on behalf of

Professor Stanton A. Glantz 

Academic Editor

PLOS ONE